# Dendritic Cells: The Long and Evolving Road towards Successful Targetability in Cancer

**DOI:** 10.3390/cells11193028

**Published:** 2022-09-27

**Authors:** Enrica Marmonti, Jacqueline Oliva-Ramirez, Cara Haymaker

**Affiliations:** Department of Translational Molecular Pathology, University of Texas MD Anderson Cancer Center, Houston, TX 77030, USA

**Keywords:** antigen-presenting cells, dendritic cells, monocytes, immunotherapy, cancer

## Abstract

Dendritic cells (DCs) are a unique myeloid cell lineage that play a central role in the priming of the adaptive immune response. As such, they are an attractive target for immune oncology based therapeutic approaches. However, targeting these cells has proven challenging with many studies proving inconclusive or of no benefit in a clinical trial setting. In this review, we highlight the known and unknown about this rare but powerful immune cell. As technologies have expanded our understanding of the complexity of DC development, subsets and response features, we are now left to apply this knowledge to the design of new therapeutic strategies in cancer. We propose that utilization of these technologies through a multiomics approach will allow for an improved directed targeting of DCs in a clinical trial setting. In addition, the DC research community should consider a consensus on subset nomenclature to distinguish new subsets from functional or phenotypic changes in response to their environment.

## 1. Introduction

Poor responses to current immunotherapies have driven the scientific community to deeply interrogate the tumor microenvironment (TME) in order to identify potential targets whose modulation significantly impacts the fate of the anti-tumor response. Recently, the spotlight has turned to dendritic cells (DCs) that, being potent antigen-presenting cells (APCs), are considered central mediators of the TME. DCs are crucial sentinel cells able to recognize diverse tumor-associated antigens (TAA).

Despite DCs critical “mentoring” role for T cells, single-agent DC-based therapies have been minimally successful and only their combination with standard of care therapies and novel immunotherapies have shown to be efficacious. The immunosuppressive TME is likely the main reason for this reduced efficacy. The TME significantly shapes the phenotype and function of DCs rendering them dysfunctional and tolerogenic in orchestrating an effective anti-tumor response. Nevertheless, this is not the exclusive reason. The knowledge circulated within the scientific community regarding DC origin, phenotype, function, differentiation and classification, not only within the TME but even in steady-state conditions, is superficial and a significant contributing factor that has led to a weak therapeutic design in targeting these cells. The interpretation of the published literature regarding the DC in tumor and steady-state healthy conditions is a minefield. Although many distinct phenotypes and subsets of DCs have been identified in tissue and circulation, a unified and standardized nomenclature system does not exist. The advent of multi-color flow cytometry and new genomic technologies (i.e., RNA seq, scRNA), have significantly refined functional and phenotypic exploration, but they have simultaneously added complexity. New subsets with intermediate phenotypes have been introduced whose origin is unclear. Frequently different nomenclature is given without verifying whether the cell type represents a distinct maturation stage of an already known cell type. To further complicate the scenario, each tissue harbors its own unique repertoire of DC subsets, which undergo significant changes during pathological conditions. As a result, open questions remain: (1) are there truly several DC subsets or we are analyzing differentiation stages? (2) Are these subsets present in normal conditions or does the microenvironment force the expression of phenotypes depending on the age, gender, metabolic state, health/disease? (3) Can we manipulate the immune response by selecting a specific DC subset or will the subsets respond equally?

The path towards successful DC targetability in cancer includes: (1) a deep characterization of tissue-specific ontogeny, phenotype and function of DC subsets at steady states; and (2) the definition of a unified and universally recognized DC classification system. Only when these two requisites are satisfied, will we be able to study and fully interpret DC subset behavior within the TME. An understanding of the mechanisms and players involved in inducing DC suppression, heterogeneity and function, both in circulation and within the TME, on the ability to orchestrate anti-tumor immunity among different cancer types and disease stages are critical points for the realization of successful cancer therapy. This knowledge may also facilitate the design of DC vaccines and the identification of predictive biomarkers of clinical outcome and treatment response. The incorporation of high-dimensional multi-Omics approaches into translational research projects, especially as a part of clinical trials, will certainly shed more light on dimensional aspects of tumor-infiltrating DC identity, better enabling the accomplishment of these goals.

In this review, we aim to discuss the latest regarding the advances in DC origin, basic features of DC biology, heterogeneity, function and targetability in homeostasis and cancer.

## 2. Distinct Origin and Development of DCs

Dendritic cells are professional APCs considered central players in the orchestration and priming of both innate and adaptive immune responses against invading or threatening pathogenic agents [1,2]. Over the years, the independent origin of DCs from monocytes and macrophages has been matter of study and intense debate. In the 1990s, many groups reported the potentiality of monocytes to differentiate in vitro into DCs upon cytokine stimulation, like granulocyte/macrophage colony-stimulating factor (GM-CSF), tumor necrosis factor-α (TNFα) and IL-4 [3]. Due to their rarity in human peripheral tissue and blood, in vitro differentiation of DCs from monocytes has been a tremendously helpful tool in gaining insights into their biology. However, multiple studies have confirmed that DCs can originate from lymphoid and non-lymphoid tissues. Monocytes have been identified as precursors of peripheral non-lymphoid organ DCs and migratory DCs during inflammation, named the monocyte-derived DC (moDC) [4,5,6,7]. Therefore, until recently, monocytes have been considered progenitors of DCs, until a DC-restricted progenitor, coined the common-DC progenitor (CDP) was discovered [8].

Myeloid cells arise during hematopoiesis from hematopoietic stem cells (HSC) which are characterized by the expression of a transmembrane glycoprotein, CD34, and lacking lineage markers (Lin^−^) [9]. Other multipotent progenitors (MMPs) or multilymphoid progenitors (MLPs) are defined as Lin^−^CD34^+^CD38^−^ and can be found in bone marrow and umbilical cord blood. In the fetal liver, there is a constant ratio of CD34^+^CD38^−^ stem cell and CD34^+^CD38^+^ progenitors, with abundant oligopotency activity [10]. By contrast, adult bone marrow is comprised predominantly of uni-lineage progenitors; primarily myeloid and erythroid, with an absence of oligopotent intermediates and few multipotent stem cells [10]. A recent study performed by Karamitros et al. showed that the MLP produce primarily B, NK and T cells as well as residual monocytes. The lymphoid primed multi-potential progenitor (LMPP) have lymphoid and myeloid potential in adults, but interestingly in vivo produce mainly myeloid cells [11]. In the development of the myeloid compartment, the main transcription factor (TF) is purine rich box 1 (PU.1), a family member of the erythroblast transformation specific (ETS) TF. The degree of PU.1 expression has been reported in the maturation of DCs, macrophages and neutrophils [12,13] and plays a crucial role in the gene expression of costimulatory molecules CD80, CD86 [14], OX40L [15] and fms-like tyrosine kinase 3 (Flt3) [16]. Additionally, cytokine receptor genes like IL-7Rα, M-CSFR, G-CSFR, G-MCSFRα are regulated by this TF [17].

This is distinctly different from monocyte development where, monocytes are produced by committed monocyte progenitors, GMP, monocyte-DC progenitors (MDPs), and in some cases from splenic reservoirs [18]. Downstream of the common monocyte progenitor (cMoP), a population of GMP, identified by the expression of CD123, CD45RA, CD135 (FLT3) [19], as well as expressing CLEC12Ahi and CD64hi in umbilical cord blood and bone marrow; can rise to pre-monocytes and mature monocytes [20]. In this commitment, the transcriptional factor for granulocytes development C/EBPα must be inhibited by PU.1 and interferon regulatory factor 8 (IRF8) to promote the induction of Kruppel-like factor 4 (Klf4), the main TF for monocyte differentiation [21,22,23]. The TF ReIB, a member of the NF-κB family, forms heterodimers with inhibitory proteins IkBs-like. Its inhibition can arrest monopoiesis and interstitial DCs development. Therefore, this process can induce the presence of monocyte precursor intermediates that promote the differentiation of some DC subsets [24].

Conversely DC TFs and development may change between the subpopulations, as further described in Table 1. They differentiate from cDCs-precursors on based on key transcription factors, like BATF3 (Basic Leucine Zipper ATF-Like Transcription Factor 3), IRF8, ID2 (DNA binding 2), ZFBTB46 (Zinc Finger and BTB Domain Containing 46) [25,26,27,28], the growth factors FLT3L and granulocyte-macrophage colony-stimulating factor (GM-CSF) [29,30,31]. Notch and PU.1 are important transcriptional factors modulating their differentiation and maturation. Specifically, PU.1 is involved in the induction of Flt3 receptor [32] and in the discrimination of the classical DC1 (cDC1) differentiation pathway [33].

### Monocytes

Interestingly, monocytes are also a heterogenous group first subdivided by Ziegler-Heitbrock et al. into 3 main subsets by the expression of CD14, a TLR-4 co-receptor for LPS, and CD16 (FcγR-III): CD14hiCD16^−^ (classical), CD14hiCD16^+^ (intermediate) and CD14loCD16^+^ (non-classical) monocytes [34,35,36,37]. Classical monocytes represent the most abundant population comprising ~85% of total peripheral blood monocytes, show high phagocytic activity and give rise to the other monocyte subpopulations [38]. M-CSF is associated with non-classical monocytes transition; hence the blockade of this pathway depletes this population [39]. During the transition from classical to intermediate and non-classical subsets the life span expands from 1.6 days to 4.3 and 7.4 days, respectively [38]. The intermediate population is ~5% of the total monocytes in the peripheral blood and is not a homogeneous group but can be subdivided based upon the expression of the Tie2/TEK angiopoietin receptor with a ratio of 35–75% Tie2^+^ cells displaying angiogenic properties [40]. Recent work reported by Villani et al., confirmed the heterogeneity of this subpopulation through single cell sequencing identifying two smaller clusters within the intermediate monocyte subset named Mono3 and Mono4. The Mono3 subset expresses genes related to trafficking, cell cycle and differentiation regulation, while Mono4 is related to cytotoxic activity due to the expression of NK genes [41]. Finally, the non-classical monocyte subset is smaller in size and represents around 10% of the monocyte population in circulation and displays opposite activities compared to the classical population [35,36].

LPS stimulation of classical and non-classical monocytes showed differing behavior in the production of IL-10, IL-6, CCL2 and G-CSF with the classical subset having the highest production. Additionally, classical monocytes can express CCR2, CXCR1, CXCR2, CLEC4D and IL-13Rα1 [42]. Therefore, the principal functions of classical monocytes are related to tissue repair, phagocytosis-defense, coagulation and apoptotic clearance due to the presence of C-type lectin and scavenger receptors [43]. The intermediate population exhibit an activated phenotype with the expression of CCR5, CD11b, CD1d, CD163, MHC class II, CLEC10A, GFRa2, the highest antigen presentation, IL-2, IFNγ and ROS production [44]. Non-classical monocytes can be defined as inflammatory monocytes with the ability to produce higher amounts of TNF-α, IL-1β, have higher expression of CD115, CD294, Siglec10 and cytoskeleton rearrangement genes compared with the other subsets [42]. Thus, they are characterized by their high migratory functions, are related to immunosurveillance, endothelium transmigration and homing to lymph nodes [45].

## 3. DCs: The Most Potent APC of the Body

DCs are fundamental in maintaining immune homeostasis and in generating peripheral and central tolerance [46,47]. Migration through the body is a critical feature of DCs’ immunological function and maturity [48]. The migration and consequent correct localization of DCs in peripheral tissues is strictly modulated by a complex regulatory network of chemotactic, non-chemotactic signals and receptors [49,50,51]. Immature DCs (iDCs) are characterized by a lower expression of major histocompatibility complex (MHC) I and II, T cell co-stimulatory ligands (e.g, CD80, CD86, CD83), adhesion molecules and cytokines (e.g., IL-12, IL-10, and TNF) [52]. In spite of their insufficient migratory and cytokine secretory ability, iDCs are fully equipped with highly active endocytic machinery for the sampling of foreign antigens [53]. Once iDCs recognize an antigen, they start the maturation process by switching their immature endocytic activity to migratory. This program induces cytoskeletal remodeling with the consequent acquisition of a faster migratory capacity, increased expression of chemokine receptors such as CCR7 for homing and secretory capability [49,54,55,56,57,58]. In contrast, mature DCs (mDCs) reside mostly in the secondary lymphoid organs where they act as APCs through their long dendrites and multiple pseudopodia [59]. In addition, they change their phenotype throughout the gradual loss of progenitor markers like CD2, CD4, CD13, CD16, CD32 and CD33, up-regulation of adhesion molecules, costimulatory molecules and MHC on their surface. The expression of CD80, but not CD86, is the best maturation marker since it is almost absent in blood precursors and appears at more mature stages [60].

The uptake of a foreign antigen represents the triggering event of the DC maturation process. DCs interact with their microenvironment through pattern recognition receptors (PPRs), induce cytokines and internalization signaling pathways [61]. The existence of diverse DC subsets characterized by a distinct repertoire of PRRs suggests a division of labor and a specialized ability to recognize and orchestrate the stimulus-specific response [30]. In order to induce an adaptive immune response, the internalized antigen needs to be processed and presented to activate cognate naïve T cells [62]. The efficiency of antigen processing is a critical requisite for the strength of the subsequent T cell response [63]. During this immunological synapsis process, costimulatory signaling (signal 2) is one of the most important features for the correct activation of T cells. The nature of the costimulatory molecule dictates the activation and inhibition of the synapsis. This will be discussed in more detail below for individual subsets and within the tumor microenvironment.

Although an extensive description of co-stimulatory molecules has been performed; the role of inhibitory receptors in regulating DC activity is still under exploration [64]. These inhibitory pathways; also defined as immune checkpoints; are physiologically critical for maintaining self-tolerance; immune homeostasis and preventing tissue damages induced by an excessive inflammatory response [65,66]. Mapping the expression of these inhibitory receptors at steady state and upon different pathological conditions provides a better understanding of DCs functionality and novel insights into the comprehension of DC heterogeneity and therapy applications.

## 4. Identification of New DC Subsets

The ability to orchestrate different immune responses against a wide range of danger signals and to interact with specific T cells has been partially attributed to the presence of functionally specialized DCs subsets [30]. The DC family is a heterogeneous composition of cell types, each with a unique origin, growth factor requirement, migration pattern and immunological function [67]. In peripheral organs, they can differentiate into specific DC subtypes based upon the expression of subset-specific transcription factors [68]. The limited understanding of their phenotypic and functional heterogeneity among tissue has significantly restricted their targetability in immunotherapy [69]. Recent findings reported important differences in the expression of activator and inhibitory molecules among DCs subsets (e.g., Programmed cell dead ligand 1; PD-L1 and T cell Ig domain and mucin domain 3; Tim3), that can further bring insights in the complexity of DC heterogeneity [64].

Previously, there were no standardized guidelines for the appropriate classification of DCs into different subtypes. The low frequency of DC subsets in tissues, rarity in blood and the laborious isolation process has significantly delayed their functional characterization in humans. Initial classifications proposed were performed based upon origin (myeloid lineage or lymphoid lineage), expression of surface markers, tissue localization (migratory DCs or tissue resident DCs), maturity (iDCs or mDCs), functionality and cytokine profiles [70]. In 2008, the first attempt from a group of experts from the International Union of Immunological Societies and the World Health Organization was made to standardize the nomenclature of different DCs subtypes in the blood [35]. DCs were categorized in two subtypes: Myeloid DCs and plasmacytoid (pDCs). Myeloid DCs were further divided into CD141^+^ (also known as thrombomodulin) and CD1c^+^ (also known BDCA1) subtypes [35]. In 2014, Guilliams et al. identified four subtypes based upon ontogeny and location, function and phenotype: Classical type I DCs (cDC1s) for CD8a^+^ and CD103^+^ DCs; Classical type 2 DCs (cDC2s) for CD11b^+^ and CD172a^+^ DCs, plasmacytoid DCs (pDCs) and monocyte-derived DCs (mo-DCs) [4].

More recently, the advent of multi-color flow cytometry and mass cytometry techniques, together with advances in transcriptomic and proteomic profiling techniques, have significantly advanced our understanding of DC biology [71,72,73]. These integrated approaches have contributed to the review of the “old” classification system and the development of a “new” extended nomenclature [30,74]. The most recent refined classification system arises from Villani et al. based upon transcriptomic profiling [41]. They confirmed the earliest classification of DCs from 2014 into four main groups (cDC1, cDC2, pDC and mo-DCs), albeit with the introduction of additional levels of heterogeneity, mainly in pDCs (named DC6) and cDC2 subsets, and the identification of new populations (AS DCs, DC4). Additional transcriptomic and proteomic studies have confirmed the Villani model, although some disagreement exists regarding some new classified populations (e.g., DC4) [75,76,77,78]. Below, we highlight DC populations identified to date. These subsets along with relevant functional and phenotypic characterizations and associated references are shown in Figure 1 and Table 1.

**Table 1 cells-11-03028-t001:** Described functions of DC subpopulations in normal conditions.

DC Subpopulation	Localization	ID Markers	Maturation Molecules Activation/Inhibition	Ontogeny and Function	References
** *Classical Type I DC (cDC1) or DC1* **	<0.01% of CD45^+^ cells in the blood of healthy donors. <0.1% of CD45^+^ cells in tissues. Blood Lymph node paracortex Other tissues (e.g., tonsil, spleen, skin, lung, intestine, ileum, payers’ patch, liver, and lymph nodes).	CD141^+^ Low/moderate CD11c, CD11b and SIRPα (Signal Regulatory Protein Alpha, CD172) CD103^+^ (α_E_ integrin) CD1c^−^ in circulation CD1c^+/−^ in tissues (e.g., skin and lungs) CD59^+^ SLAMF8^+^ CD26^+^ CD8α^+^ Negative for moDCs markers: CD14, CD16, CD209 and SIRP CD45RA^−^ CD123^−^	DEC-205^+^ (dendritic and epithelial cells-205) XCR1^+^ (the chemokine receptor X-C Motif Chemokine Receptor 1) *maturation marker* CLEC9A^+^ (C type lectin receptor Clec9A, CD370) Axl^+^ receptor tyrosine kinase CADM1^+^ (Cell adhesion molecule 1) CD135^+^ (Flt3L receptor) High expression of MHCII IDO1/IDO2 (Indoleamine 2,3 Dioxygenase 1 and 2 TLR-3 and TLR-9 BTLA (B and T Lymphocyte Attenuator, CD272) Tim3 and PD-L1 inhibitory receptors Higher CD40 and lower CD86 expression than cDC2s	Relatively homogenous population in comparison to cDC2. Polarize activated CD4^+^ T cells toward Th1 and away from Th2 phenotype. Major source of IL-12 in vivo. Immune responses against cancer and pathogens infection. CLEC9^+^ CD141^+^ cells are producers of IFNγ and CD8^+^ T cell activation by cross presentation in response to TLR-3 ligation.	[41,64,78,79,80,81,82,83,84,85,86,87,88,89,90,91,92,93,94,95,96]
** *Classical Type 2 DC (cDC2)* **	Lymph nodes in subcapsular sinus. Two distinct subsets based upon CD5 expression CD5^high^ and CD5^low^ in blood and skin.	CD11b^+^ CD11c^+^ CD33^+^ CD13^+^ CD172α^+^ Discriminatory marker CD1c (BDCA1) CD5high DCs express high levels of cDC2-specific genes, while CD5low DCs preferentially express monocytes-related genes. A population with analogous phenotype CD1c^+^CD14^+^CD5^low^ cDCs (VCAM, FCN1, S100A8, S100A9) was named DC3. Their ontogeny and functions remain unknown.	Mgl2 or CLEC12A, CD2 FCεR1 SIRPα CCR2 CCR6 TLR2, TLR4–6, TLR8 and TLR9 Dependent on IRF4, IRF2, TRAF6, KLF2, RelB, RBP-J transcription factors and Notch signaling. The CD5^high^ cells have strong IRF4, CCR7, CD207, TLR3 expression. Additional markers: CLEC10A VEGFA, FC R2A.	Secretion of inflammatory cytokines: TNF-α, IL-12p70, IL-23, TNF-α, IL-1, IL-6, IL-8, IL-12, IL-18 and chemokines: CCL3, CCL4 and CXCL8. Uptake of exogenous antigens, specifically tumor antigen. T immune responses against extracellular bacterial and fungal pathogens. Polarization of Th17, Th2, Th1, Th22, Treg and cytotoxic responses. By the induction of Th2 responses are involved in humoral responses via IL-6 and IL-1β secretion. Higher potential to induce CD4^+^ T cell proliferation than cDC1. CD5^high^ cells induce IL-10 producing Treg cells, whereas CD5^low^ cells induce IFNγ producing T cells.	[31,33,41,64,88,90,97,98,99,100,101,102,103,104,105,106,107,108,109,110,111,112]
** *DC2* **	DC2Bs have been detected in the spleen apparently are absent in circulation.	CD1c^+^ Compared to DC3 have slightly higher expression of: MHCII CD1c CD11c CD5 Unique expression of CD32b CD301 (macrophage galactose-type C-type lectin, CLEC10A) FC R1A (the alpha chain of the high affinity receptor for IgE). Two subpopulations by CLEC10A expression: **DC2A**: CD1^low^ CLEC10A^−^ CLECL4^high^ **DC2B**: CD1c^+^ CLEC10A^+^ CLEC4A^low^	TFs, gene associated with lipid antigen presentation and metabolism (CD1E, NPC2, PSAP) have found to be enriched in DC2Bs, whereas CD3E in DC2A subset.	DC2 cells show greater secretory ability of CCL19, IL-10, IL-12B and IL-18 than DC3 cells. DC2B exhibited pro-inflammatory potential with an increased expression of IL-1B. DC2A have anti-inflammatory phenotype characterized by higher levels of transcript for amphiregulin (AREG), IDO1, the immunomodulatory receptor CD300a and IL-22 binding protein.	[41,78,98,113,114]
** *DC3* **	Expanded DC3 populations have been observed in blood of patients with systemic lupus erythematosus (LE) and in skin of patients with psoriasis	CD1c^+^ CD14 ^low/high^ CD5^low^ Gene signature (CD14, S100A9, S100A8). Unique expression of CD163 (a scavenger receptor and PPR for bacterial) and CD36.	CD88^−^ BTLA^−^ Additional key markers VCAM, LYZ and ANXA1 Expresses unique markers enriched for antigen processing, MHCII and leukocytes activation. Express CCR7, co-stimulatory molecules (CD80, CD86, CD70, CD40), T cell attracting chemokines (CCL5, CCL19, CCL17, CCL22, CXCL9, CXCL10, CXCL11 and CXCL13).	DC3 cells are characterized by acute and chronic inflammatory gene signature. DC3 function in vivo and its contribution to pathological conditions are still uncertain. Produce IL1B and IL23A during pathogenesis. Produce high amount of IL-12p70, IL-13, IL-17 and IL-10, TNF-α and IL-1β, CCL2, CCL1, CCL3, CXCL1, CXCL3 and CXCL5. DC3s induce T_RM_ differentiation from naïve CD8^+^T cells.	[41,73,75,78,100,112,115]
** *CD16^+^ DC or DC4* **		Lin^−^ HLA-DR^+^ CD11c^+^ CD14^low/high^ CD16^+^ CD141^−^ CD1c^−^ High expression of PPP1R14A and DAB2	Slan^+^ cells? CD85d (ILT-4) and CD85h (ILT-1) CD115 CD31 SLC7A7 CD98 Siglec-10	Gene of sets related to type I interferon signaling and virus response. IFN-α/β signaling? Further studies are needed to clarify the phenotype and function of this newly described subset.	[41,98,116,117,118]
** *AXL^+^ SIGLEC6^+^ DC (AS DC) or DC5* **	2–3% of all DC compartment Two subsets identified: CD123^+^ CD11c^−^ AS DCs (0.7%, of the CD45^+^ HLADR^+^ fraction) CD123^−/low^ CD11c^+^ AS DCs (1.7% of the CD45^+^ HLADR^+^ fraction) Found in secondary lymphoid organs, like tonsils, co-localizing with T cells in situ.	AXL^+^ SIGLEC6^+^ SIGLEC1^+^ SIGLEC2^+^ The commonly markers used to identify pDCs (CD123, CD303 and CD304, CD123, CD11c) have also been found. The identity of AS DCs has not been clarified given their heterogeneity, specifically with a spectral expression for CD123, CD5, or CD11c.	CD221 and CD169, CD39 and IRAP AXL^+^ SIGLEC6^+^ DCscan differentiate into cDC2 showing a progenitor potential for cDCs.	As DCs do not secrete IFNα, induce T cell activation (like cDCs), Treg formation or B cell activation. Both CD123^+^ CD11c^−^ AS DCs and CD123^−/low^ CD11c^+^ AS DCs subsets are potent stimulators of allogenic CD4^+^ and CD8^+^ T cell proliferation.	[41,71,77,114,117,119]
** *pDC or DC6* **	0.2–0.4% of total CD45^+^ circulating cells.	Lin^−^ CD11c^−^ CD11b^−^ CD13^−^ CD123/IL3R^+^ CD303/BDCA2^+^ CD304/BDCA4^+^ Used as a discriminative lineage marker to identify pDCs: ILT7 (immunoglobulin-like transcript 7) CD4 BCL11A TCF4 The co-staining of CD123 with HLA-DR is essential to exclude HLA-DR-granulocytic populations expressing CD123. CD68 is a marker useful for distinguishing them from DC1 and DC2. Named by Villani et al. as DC6, pDC differentiation is modulated by IRF8, IRF7, E2.2, Runx2, SpiB, IRF4, BCL11A and ZEB2 expression. Dual myeloid-lymphoid origin	At steady state, pDCs exhibits low levels of MHC class I and II, and low to undetectable level of costimulatory molecules CD40, CD80 and CD86. PD-L1 is negligible on their surface and very low expression of Tim3 is present at basal and stimulatory conditions. After stimulation, pDCs express IFN-α/β and differentiate into mature MHC class I/IIhigh, CD80^+^, CD40^+^ CD86^+^. TLR7 and TLR9 CCR2, CCR5, CXCR3 and CXCR4 chemokine receptors has been also detected	Natural IFN-α producing cells in response to virus, IL-3, and bacterial components. Moderate amounts of TNF-α and IL-6 after viral stimulation. Signaling pathway for CLEC4C, NRP1 and IL3RA. Co-stimulatory and activation of NK cell, cDCs, T cells and B cells. Tolerogenic functions by inducing T cell depletion, CD4 T cell anergy and T reg differentiation. Cross-priming post-antigen presentation by driving antigen transfer to bystander DCs through pDCs-derived exosomes.	[41,64,88,114,120,121,122,123,124,125,126,127,128,129,130,131,132,133,134]
** *Monocyte-derived DC* **		HLA-DR^+^ CD11^+^ CD1c^+^ CD1a^+^ CD1b^+^ FcεRI^+^ Negative for: CD16, CD163, CD206, CD209, CD14 and CD11b. Recently described LSP-1 as a marker distinguishing from monocytes, macrophages, cDCs or DC3	Mainly express CCR2 and CXC3CR1.	Inflammatory DCs or Tip (TNFα and inducible nitric oxide synthase producing). Produce IL-6, IL-23, IL-1β and TNFα upon activation. Bacterial phagocytosis, iNOS-dependent bacterial killing and tissue toxicity. Depending on the inflammatory stimulus, they induce T helper (Th1), Th17 or Th2 responses. Induce CD4^+^ T proliferation and cross-present to CD8^+^ T cells.	[31,38,88,135,136,137,138,139,140,141,142,143,144,145,146,147,148].
** *mregDC* **	This population has been found in inflammatory environments, including solid tumors and Crohn’s disease lesions.	This population has been defined as CCR7^+^ DCs, or “activated DCs” or “mature DCs enriched in immunoregulatory molecules” (mregDCs) with the expression of LAMP3. Regulatory markers CD274, CD200, FAS and ALDH1A2	Low levels of TLR signaling genes and increased levels of migratory genes the expression of maturation markers (CCR7, CD40, RELB and CD83)	Modulating Th2 response due to the expression of IL4R, IL4I1, CCL17, CCL22 and BCL2L1 genes. In human and mouse non- small cell lung cancers, the mregDC program is expressed by canonical DC1s and DC2s upon uptake of tumor antigens. The role of this new subset is still under exploration both in homeostatic and pathological conditions.	[41,98,122,140,149,150]

## 5. DC Crosstalk within the Tumor Microenvironment

Tumor cells have developed strategies to immune-evade or tolerate the anti-tumor activity of DCs; however, their infiltration in the TME is an essential requisite for the orchestration of efficacious immune surveillance and tumor eradication [151,152]. Although low in frequency, tumor-infiltrating DCs (TIDCs) have been detected in a variety of tumor types [153,154]. Interestingly a shift in DC phenotype from immunostimulant to immunosuppressive or tolerogenic occurs inside the TME. These immunosuppressive cells are generically termed regulatory DCs (DC regs) and include various subsets with different states of maturity, whose normal function is undermined by tumor-derived signals [155]. Complex multi-factors derived from the tumor microenvironment, including VEGF, non-classical HLA class I, death ligands (FasL and TRAIL), pro-and-anti-inflammatory cytokines (TNF-α, IL-10, IL-1β, TGFβ, IL-8) and metabolites (e.g., IDO, ROS, RNS, NO), are directly responsible for DCs functional transformation [156,157,158,159,160].

While it is still not clear if DC presence in the TME is associated with a poor or good prognosis in different tumor types, DC dysfunction directly translates into aberrations for T cell activation [161]. This can be due to heterogeneity in their phenotype and function, the biology of tumor types and methods of identification used [162,163,164,165]. Different studies demonstrated how the type, phenotype and frequency of DCs inside the tumor dynamically change throughout disease progression [166]. The frequency of circulating and infiltrating DCs are generally lower in patients with cancer in comparison to normal patients [167,168]. This alteration is associated with poor therapeutic outcome, survival and restricted success of DC-based vaccine therapies [168,169,170,171]. TIDCs exhibit a distinct infiltration and density pattern based on their maturation status [172]. Mature DCs are mainly detected at the peritumoral zone of tumor lesion compared to immature DCs, that are retained at the tumor parenchyma [173] and are characterized by the inability to migrate out from the internal compartment [174,175] cifically, the chemotactic axes modulating their recruitment and retention from blood to inside the tumor parenchyma are disabled. These chemokines can be secreted by the tumor itself (e.g., CCL4, CCL5, XCL1) or by other infiltrating immune cells [92,176]. Upon antigen uptake, TIDCs exhibit impaired migratory ability to home to draining lymph nodes and altered expression of costimulatory molecules. However, changes in chemokine receptor profiles on the DC surface are fluid and responsive to their microenvironment. For example, CCR7 expression levels are enhanced after the direct contact with apoptotic tumors and its presence is correlated with CD141^+^ DCs, intra-tumoral T cells, and better clinical outcomes [177,178]. Enrichment for CCR7 expression produces a dendritic cell therapy product with enhanced migratory ability able to generate a powerful anti-tumor response [179].

A pathological myelopoiesis triggered by the tumor might also explain the low frequency of DCs found in cancer patients [155]. This results in the accumulation of immature myeloid progenitors at lymphatic and tumor locations [180]. The presence of immature cells in the peripheral blood of lung cancer patients have been closely associated with an increased plasma level of VEGFA, GM-CSF, M-CSF, IL-10, and TGF-β [181]. These immunosuppressive mediators released intratumorally promote DC differentiation toward myeloid-derived suppressor cells (MDSC) or TAM [182,183,184,185]. Soluble factors secreted by human renal carcinoma tumor cells and pancreatic cancer tumor cells have been shown to inhibit DCs differentiation from CD34^+^ progenitors and trigger a lineage commitment toward CD14^+^ monocytes lacking APC function [183,186].

Comparative transcriptomic analysis has defined higher frequencies of DCs in lung cancer in tumor tissue as compared to non-tumor tissue with less expression of co-stimulatory molecules on their surface (CD40, CD80, CD86 and MHC class II) showing an association with the tumor impairment of DC activation [187,188]. Previous studies have shown how the release of IL-10 from the tumor downregulates the expression of CD40 on DCs and DC precursors, suppressing their maturation and function [189]. Low expression levels of CD40 on DCs support tumor growth, in contrast high expression levels induce tumor-regressing T cell responses [190]. The impact of the Programmed cell dead 1 and the ligand (PD-1/PD-L1) axis on DCs is a potent tumor immune evasion mechanism that can impair DC costimulatory functions and subsequent T cell priming and re-stimulation [191,192]. Mayoux et al. [193] have demonstrated that both peripheral and tumor infiltrated cDC1 and cDC2 express high PD-L1 under steady-state conditions. Recent data has shown that DCs expressing PD-L1 correlate with CD8^+^ T cell infiltration and good prognosis in colon cancer [194], and improved efficacy of anti-PD-1 treatment in ovarian cancer and melanoma [195].

Components of the MHC class I antigen processing machinery have been found to be modified in multiple solid and hematologic tumors and this modification or impairment can occur at the epigenetic, transcriptional or at post-transcriptional level [196]. Immature DCs incubated with primary oral squamous cell carcinoma cell lines show significantly lower expression of several MHC class I processing components like MB1 (β5), LMP2,7,10, and ERp57 [197], induced by tumor derived ganglioside [198] or IRF-8 [199]. These changes were translated into a decreased ability to cross-present antigen and, thus, tumor recognition. Abnormal lipids accumulation in DCs represents another essential mechanism that contribute to defective processing of TAA [200].

### 5.1. Tumor-Infiltrating DCs Subsets (TIDCs)

Recently, single cell RNA seq has deepened our characterization of TIDCs in various human tumors, including non–small cell lung cancer [141,149,201,202], head and neck squamous carcinoma (HNSCC) [203], hepatocellular carcinoma [204], melanoma [98,205], cutaneous squamous cell carcinoma [206], colorectal [201,207], ovarian, breast [201], and renal cell cancers [208]. Comparative transcriptomic analysis has defined higher frequencies of all DCs subtypes within the tumor lesion in comparison to non-tumor tissue [187,201]. Heterogeneities in immune states and cell type composition of DCs typically observed among tumor types and individual patients make it challenging to define a behavior for each individual DC subset in cancer [209,210,211,212]. Immune phenotypes of the tumor (hot versus cold), immuno-profiling assays used for TME characterization, age, gender, tumor stage, histological type, treatment history, number of treatment lines, dosing, and clinical target volume are some of the factors responsible for large variability in DC phenotype and function. Qian et al. reported different fractions of DCs clusters among three different cancer types (lung, colorectal and ovarian); however, all three were found to be highly infiltrated by cDC2s. Similarly, Michea et al. identified different qualitatively and quantitively features specific for each DC subset in function of breast-cancer subtypes (“cold” versus “hot” tumor) [165]. In addition, Laoui et al. showed as the relative proportions of distinct DCs subsets and consequent function evolve during cancer progression [138].

The integration and comparison of different scRNA-seq studies considering different tumor type is very challenging due the differences in methodological approach (tissue dissociation, bioinformatic approaches, scRNA-seq platform) and distinct nomenclature used. Gerardh et al. by performing a meta-analysis of eight currently available scRNA-seq datasets regarding tumor-infiltrating DC subsets across five human cancers revealed five existent DCs across tumors regardless the origin tissue, genetic signatures of tumor cells, or composition of the tumor microenvironment [213]. cDC1, pDCs and DC3 are highly preserved and cDC2 and cDC2/moDC less conserved, this last express markers of monocytes and macrophages (i.e., CD14, CD64 and CD163) and cDC2 (i.e., CD1c, CD206 and FcεR) that represents an intermediate translational status between CD1c^+^ CD5^+^ CD14^−^ CD88^−^ CD163^−^ cDC2 and CD88^+^ CD14^+^ CD163^+^ CD1c^−^ CD5^−^ DCs [43,103,107,213]. This population has also been identified by Zilionis et al. in lung cancer as “DC3”, by Maier et al. as “mreg DCs” and by Qian et al. as “migratory DCs” [141,149,201]. Additionally, to the cDC2/moDC (CD1c^+^CD14^+^) population, Michea et al., has identified the prevalent infiltration in breast cancer of CD1c^−^CD14^+^ cells with a monocytes/macrophage- like phenotype and CD56^+^CD14^+^ cells, with interferon-producing potentiality whose nature is still controversial [165]. Transcriptomic analysis performed by Qian et al., considering lung, colorectal, ovary and breast cancers, have highlighted the shared infiltration of an alternative cDC2 subtype with a Langerhans-like phenotype (CD1C, CD1A and CD207) [201]. In melanoma, specific features of DC subsets have been described in peripheral blood and tumor compared to healthy donors, which displayed an overall higher basal activation status [214]. Michea et al., have shown in breast cancer that pDCs express highest number of upregulated genes followed by cDC1c and CD1c^−^ CD14^+^ cells [165]. They have proposed that ontogeny, tissue imprinting, or their combined effect can explain this pDC-specific signature [165]. In addition, they found increased transcripts in breast cancer DCs subsets: in pDCs the negative regulator CD5 transcript has been detected, in cDC2s the secretoglobulins SCGB2A2/SCGB1D2 and AGR2 (a protein disulfide isomerase needed for mucin folding) transcripts; in CD14^+^ DCs the SCGB2A2 transcript; in cDC1s the TACI (TNFRSF13B), a member of the cytokine TNF receptor superfamily and CCL7 in CD1c^−^CD14^+^ DCs [165]. Similarly, Qian et al. reported a list of transcription factors that underlies each DC phenotype (BATF3 for cDC1s, CEBPB for cDC2s, NFKB2 for migratory DCs and TCF4 for pDCs), plus novel transcription factors tumor associated (for detailed see [201]). They identified BIRC3 as new marker for cDC maturation, upregulation of IFN-α/β signaling and folate metabolism in CCR7^+^ migratory DCs as a differentiated status of cDC2 but not cDC1c subset [201]. Below, we describe the current state of the field related to each of these subsets and cancer.

#### 5.1.1. Tumor Infiltrating cDC1

Representing < 0.2% of infiltrating human CD45^+^ leukocytes, CD141^+^ CD8a^+^ XCR1^+^ CLEC9A^+^ BATF3^+^ cDC1 is a rare population in human cancers [165,187,215]. In melanoma, cDC1 are less than 10% of all DCs populations detected intratumorally [123]. Single cell analysis has revealed a cDC1 transcriptomic signature in melanoma, lung adenocarcinoma and breast cancer [165,187,205,216]. Genes previously associated with cDC1, including CLEC9A, XCR1, CADM1, TBHD and IRF8 are conserved across multiple tumor types [138,141,165,187,205]. A positive correlation between higher expression of cDC1 transcript, T cell signatures and good prognosis has been highlighted in these tumors [207,217]. Contrary to cDC2s, sDC1 can deliver intact TAA to draining lymph nodes, hence making them one of the strongest DCs for priming naïve CD8^+^ cells [156,171]. Batf3, Clec9A or Wdfy4 deficiencies in DCs result in antigen cross presentation defects, reduced CD8^+^ T cell infiltration and inability to reject tumors [218,219,220]. Furthermore, the occurrence of cDC1s in the TME has been correlated with high number of infiltrating NK cells, better survival and improved clinical responses to anti-PD1 therapy in melanoma patients [221]. The frequency and function of cDC1s is reduced in patients with ovarian and prostate cancer [167]. Although no changes in cDC1 density have been observed between non-tumor and tumor tissue in melanoma patients, their decreased frequency in circulation have suggested active recruitment from blood into the tumor [123].

Low infiltration of cDC1s observed in some tumor types might be a consequence of a systemic suppression of hematopoiesis in the bone marrow, or a local inhibition by the reduced production of growth factors, like FLT3 and G-CSF which are important for differentiation, expansion and survival [138,171,222]. Patients with pancreatic cancer have low CDPs, pre-DCs and cDC1s in bone marrow and high levels of G-CSF in tumor [222]. High levels of IRF8 were detected in tumor-associated cDC1s [138] and its downregulation impairs the proper differentiation from the progenitors [222]. Alternatively, tumor intrinsic Wnt-β catenin signaling and PGE2 released COX (cyclooxygenase)-mediated are the major mechanisms driving tumor exclusion of cDC1s [221,223]. Alteration in chemokine routes modulating cDC1 recruitment intratumorally is an additional cause of their low frequency. The high presence of CXCL9, CXCL10, and CXCL11 in the TME have been associated with favorable clinical outcome in some cancers [224]. This subset is the primary source of intratumoral chemokines which directly modulate CXCR3^+^ effector T cell infiltration [169], CD8^+^ T cell cytolytic activity and IL-12-mediated IFNγ-production [225]. Indeed, cDC1 expression of the XCR1 chemokine receptor, CXCL9 and IL-12 is indispensable for breast cancer tumor rejection [226]. Additionally, NK cells directly attract cDC1 accumulation inside the tumor lesion by XCL1 and CCL5 chemokines and promote their differentiation under the release of Flt3 factor [216] with a strong inhibition of this axis induced by PGE2 tumor secretion [221]. In addition, cDC1s within the TME express high levels of CCR7 which predict T cell infiltration and improve outcome in melanoma patients [177].

IFN-α/β signaling activation in cDC1s is essential for cross-presentation and cancer immunosurveillance [226]. Studies demonstrate that the absence of the cytokine intratumorally is enough to generate an inefficient CD8^+^ T cell response [227]. cDC1s modulates anti-tumor response through production of IFNλ [228]. It has been demonstrated in melanoma that there is a higher secretion of IFNλ1 and a potent responsiveness upon TLR stimulation compared to normal tissue-infiltrated cDC1s in basal conditions. Thus, high proportions of activated IFNλ^+^ cDC1s in melanoma and breast cancer correlate with better outcome hence supporting their in vivo targetability [123,228]. The expression of inhibitory receptors can also significantly impact cDC1 function and high expression of PD-L1, TIM-3 and cytotoxic T lymphocyte antigen 4 (CTLA-4) have been observed on cDC1s within human breast cancers [229,230]. Tim3 has been shown to regulate cDC1 function and response to chemotherapy in a murine model of breast cancer [230] and its ligation can directly inhibit DC activation [229,231]. Upon Tim3 interaction with HMGB1, pro-inflammatory cytokine secretion by cDC1s is limited, thus suppressing innate immune responses [229]. Anti-Tim3 antagonistic antibodies have been demonstrated to improve the response to chemotherapy by increasing the expression of CXCR3 and its chemokine ligand CXCL9 expression by cDC1 in a model of triple negative and luminal B disease [230].

#### 5.1.2. cDC2

Despite a wide array of evidence supporting a predominant role of cDC1s in anti-tumor immunity, CD1c^+^ CD14^−^ cDC2s also participate by infiltrating tumors and triggering CD4^+^ T cell responses at tumor draining lymph nodes [102,138]. Within the tumor tissue, together with pDCs, these subsets are the most common populations, reaching a frequency of 35% in melanoma [123]. In the TME, cDC2s are characterized by high gene expression of CD1E, CD207, CD1A, CD1B and FCεR1A [141,166] and IRF8 [138].

This subset is strongly associated with a positive prognosis, their increased infiltration abundance and the presence of a cDC2 gene signature has been associated with improved progression-free survival in head and neck [102] and NSCLC cancer patients [141]. Moreover, significant correlations between high intra-tumoral density of cDC2s, low frequency of Tregs, high CD4^+^ T cells abundance represent a better outcome of HNCC and melanoma patients [102]. Surprisingly, Treg depletion significantly enhances the cDC2s ability to induce CD4^+^ T cell responses and antitumor protection [102]. They are also characterized by the ability to induce cytotoxic T cell responses, justifying their applicability in DC vaccination protocols [232] and are unique in their Th17-activation capacity related to the secretion of IL-23, IL-1β and IL-6 cytokines [138]. As demonstrated by Laoui et al., they have a higher prophylactic potential than cDC1 vaccination for tumor types enriched with MDSCs and M2-oriented TAMs whose control is less dependent on cytotoxic T cells [138].

Conversely, Cuevas et al. have reported no differences in cDC2 frequencies between non-tumor and tumor tissues nor between blood and tumor tissue in patients with melanoma. Nevertheless, they have found that cDC2s in melanoma tumors are dysfunctional, characterized by high CD80^+^ expression and production of IL-12p40/p70 at basal conditions and impaired production of TNF-α upon TLR triggering [214]. Micha et al. identified higher checkpoint genes expression like VTCN1, CEACAM6, TNFRSF14 in tumor tissue in comparison to juxta-tumoral samples [166]. All the dysfunctional features contribute to the loss of positive inter-relations with pDCs and cDC1s within the TME hence, their accumulation might correlate with poor outcome [123].

A recent study demonstrated that once injected into a human organotypic melanoma culture, cDC2 can follow two different fates: become tolerogenic or acquire the CD14^+^ monocytic marker [233]. The CD14^+^ cDC2 population, previously identified in tumor settings as cDC2/moDCs, is also detected in melanoma lesions, characterized by immunosuppressive function and higher expression of additional macrophage-like markers (CD163, CD206, MerTK) and PD-L1 in comparison to CD14^−^ cDC2s [233]. The enrichment of CD14^+^ cDC2 was also observed in other tumor types like metastatic melanoma, leukemia, breast, lung and colorectal cancers [111,138,166]. HPV-associated oropharyngeal squamous cell carcinoma is highly infiltrated with DC3s and their presence positively correlates with Tbet^+^, tumor-specific T cell infiltration and prolonged survival [234]. In luminal breast cancer primary tumors, Bourdely et al. have shown that a population of inflammatory DCs CD11c^+^ FCeRI^+^ CD14^+^ CD5^−^ CD1c^+^ completely aligned phenotypically with the DC3 and their presence correlates with the abundance of CD8^+^CD103^+^CD69^+^ tissue-resident memory T cells [100]. How these DC3s are generated and function within the TME remains yet to be determined [234].

#### 5.1.3. pDCs

CD11c^−^ CD123^+^ BDCA2^+^ pDCs are a well-represented population in the TME, varying between 30 and 76% of all TIDCs depending on the tumor model [138]. An active recruitment of pDCs from blood to the tumor has been observed in melanoma patients [123]. The population is identified within the TME by the high expression of IL3RA, CLEC4A, TLR9, GZMB, IRF7, LILRA4 and TCF4 genes [141,166]. Once activated via TLR7/9, pDCs directly participate in the anti-tumor immune response through antigen presentation, although with a weaker capability than cDC subsets [161]. Based on their ability to secrete IFN and TFN-α, this subset is considered to have remarkable antitumor therapeutic potential. In fact, pDCs are the highest producers of IFN-α/β, which inhibits tumor proliferation, angiogenesis and metastasis and directly modulates the action of other immune cell types (cDCs, NK and T cells) [212,235]. Their high tumoral infiltration has been correlated with overall survival of colon and triple negative breast cancer patients [236,237]. The administration of autologous activated pDCs loaded with tumor-associated peptides to melanoma patients has been shown to induce specific CD4^+^ and CD8^+^ T cells responses [238].

Despite that, the role of pDCs in the TME still remains controversial as many different observations are present in the literature regarding pDCs having an immunostimulatory or immunosuppressive role. Tumoral infiltration of human pDCs is also associated with tumor aggressiveness [239] and poor prognosis across multiple tumor types, like breast, ovarian and oral cancer [240,241,242]. Their intratumoral accumulation contributes to breast cancer lymph node metastasis via the CXCR4/SDF-1 axis [239,243]. Key hallmarks of tolerogenic pDCs include poor maturation and reduced IFN-α/β secretory capacity. This immunosuppressive phenotype has been associated with the leukocyte upregulation of immunoglobulin-like receptor (LILR) family genes [244], granzyme B production [245], and loss of CD86, CD83, CD80 and LAMP3 expression [246]. In human melanoma and head and neck squamous cell carcinoma (HNSCC), pDCs are actively recruited into the TME and their high infiltration confers a poor prognosis [123,247]. They exhibit diminished ability in producing IFN-α upon CpG-oligonucleotide stimulation [248] which favors Treg expansion and cancer progression [123,249]. Different mechanisms have been related to these phenomena, including TLR downregulation, secretion of tumor-derived soluble factors and upregulation of inhibitory receptors [250]. Additionally, decreased IFN-α secretion upon TLR-7 and TLR-9 stimulation have been reported in breast and ovarian cancers [212,239,249], probably due to the relocation of TLRs to late endosomal compartments [251]. A pDC population marked by high expression of LAG3 has been identified in tumor-invaded lymph nodes and skin metastasis featured by a limited secretory IFN-α ability and enhanced IL-6 production [252].

The induction of Tregs through inducible T cell co-stimulator and its ligand (ICOS/ICOSL), OX40/OX40L pathways and indoleamine 2,3-dioxygenase (IDO) expression are considered the most relevant pro-tumorigenic effects of pDCs [249,252,253,254]. A direct contribution of pDCs through ICOSL upregulation to tumor progression has been reported in melanoma, breast cancer, ovarian cancer and liver tumors [255]. ICOS-L^+^ pDC tumor infiltration is associated with poor prognosis and disease progression in both breast and ovarian cancer patients [253,256]. In addition, the co-stimulation of OX40L on pDCs also supports melanoma progression by promoting Th2 polarization and regulatory immunity [255]. Conversely, a pDC subset expressing OX40hi ICOSLlo/null has been found in HSCC that enhances the cDCs antigen-specific priming of CD8^+^ T cells [257]. IDO1 expression can be driven by tumor PGE2 [258], and it has been defined as a new useful prognostic marker at early stages of the disease [259]. Other inhibitory receptors involved in T cell tolerance might exert an essential role in their pro-tumorigenic activity. High expression of CD86 on pDCs in patients with chronic myeloid leukemia has been defined as a strong predictor for disease recurrence [260]. Additionally, their high expression of PD-L1 in multiple myeloma is responsible for immune dysfunction and tumoral immune-escape [261]. More studies are needed to identify pDCs diversity in cancer and their exact competence in modulating cancer progression, metastasis and immune regulation [262].

#### 5.1.4. mregDCs

This unique cluster of DCs, enriched in immunoregulatory (CD200, PD-L1, CD274) and maturation molecules (MHC-II, CD40, IL12, CD86, PD-L1, PD-L2), was identified by Maier et al. in human and murine non-small lung cancers [149]. They are exclusively found in tissues without a counterpart in the blood. Their uncertain nomenclature in tumors resulted in their classification as DC3 [213], LAMP3^+^ DCs [204], CCR7^+^ DCs [201], or BATF3^+^ DCs [207]. A higher frequency of mregDCs have been found inside tumor lesions in comparison to normal tissue and shown to be correlated with improved patient survival in NSCLC and colorectal cancer [141,207]. Although tumor-infiltrating mregDCs express key cDC1s and cDC2s markers (e.g., XCR1L, CD103 or CD11b), they are distinguished from them by higher CCR7 expression [75,149]. Some of the most highly expressed genes across tumor types are CCL19, CD40, BIRC3, LAMP3, LY75 (see review [213] for details).

CITE-seq (cellular indexing of transcriptomes and epitopes by sequencing) analysis confirms the genomics analogy between cDCs and mregDCs, suggesting a possible differentiative potential of cDC1s and cDC2s into mregDCs. A tumor driven program seems to directly modulate the acquisition of mreg DC phenotype [149], depicted by low levels of TLR signaling genes and increased levels of migratory genes. The uptake of apoptotic cells via receptor kinase molecule Axl is crucial for the induction of the mregDC program in cDC1. Within tumors, mreg DCs stimulate antitumor CD8^+^ T cells or NK cells through IL-12 production upon sensing IFN-γ released by neighboring T or NK cells [149,263,264]. Since their expression of Th2 response genes like IL-4R, IL-4L1, CCL17, CCL22 and BCL2L1, mreg DCs might also activate tumor-specific CD4^+^ T cell responses (for details see [149]). Direct stimulation of NF-κB pathway amplifies mregDCs infiltration and activation [263]. Tumor-infiltrating mregDCs highly express PD-L1, whose upregulation is induced by the phagocytic receptor Axl. They are also characterized by the secretion of IL-4, the most predominant cytokine in resistant tumors, that suppresses IL-12 production. Blockage of IL-4 on murine mregDC1 have shown to restore IFNγ-mediated IL-12 production and cause reduction in tumor growth [149].

#### 5.1.5. moDCs

HLA-DR^+^ CD11c^+^ CD14^+^ moDCs represent a small fraction of the TME infiltrate that arise from differentiation from the monocytic compartment [265]. Neutralization of the chemokine CCL2 or inhibition of colony-stimulating factor-1 (CSF-1) receptor signaling prevent monocyte infiltration into the lymph node or tumor lesions, with a consequent reduced recruitment of tumor-specific T cells and damped antitumor responses [265]. Single-cell RNA sequencing data from patients with metastatic melanoma treated with anti-PD1 have confirmed the presence of a bimodal differentiation trajectory of tumor-infiltrating monocytes into moDCs or macrophages. CSF1R-expressing myeloid cells can differentiate toward TAMs with increasing expression of MARCO, CD163 and APOE, or toward moDCs with increasing expression of CLEC10A, HLA-DR, CST7 and CD1C [266]. MoDCs are significantly more abundant in responders to anti-PD1 therapy compared to non-responders. These data suggest the crucial role of monocyte-derived APCs in modulating the response to PD-1 checkpoint blockade and providing a therapeutic target for combination therapy [266].

Polarization of monocytes towards the moDCs phenotype within the TME is associated with the acquisition of IL-15 production to induce Th1 responses [267] or with the expression of TRAIL to mediate tumor cell apoptosis [268]. In addition to exerting direct tumoricidal function via iNO production, moDCs can also serve as antigen-presenting cells and mediate effector functions via Th1 promoting cytokine production (TNFα and IL-12) [138,265]. Despite that, tumor-associated moDCs are less efficient in activating naïve antigen-specific T cells [138]. Moreover, these cells produce the highest levels of the inflammatory cytokines (TNFα, IL-6 and IL-1β), monocyte and neutrophil attracting chemokines (CCL2, CCL4 and CXCL1) and reactive oxygen species (iNOS) [138,265]. Instead of migrating to sentinel lymph nodes, they are required to license the cytotoxic activity of CD8^+^ T cells in situ [166]. MoDCs lack lymph node migratory capacity since they do not express CCR7 on their surface [138]. Transcriptomic analysis revealed moDC accumulation in multiple tumor types (e.g, breast, lung and colorectal cancers) [141,166,269] that was also correlated with CD8^+^ T cell activation and treatment response [265].

An impaired in vitro differentiation of colorectal cancer patient monocytes into immature DCs has been observed compared to those from healthy donors which appears more altered during advanced stages of the disease [270]. This is a consequence of the tumor influence in promoting altered maturation and early apoptosis of moDCs [271]. In multiple murine sarcoma models, TME release of retinoic acid polarizes intertumoral monocyte differentiation toward TAMs by downregulation of IRF4 [272]. Alternatively, melanoma-secreted lysosomes induce moDC apoptosis and limit cancer immunotherapy [273]. In addition, moDCs from multiple myeloma patients are defective in migration and autocrine secretion ability [274].

### 5.2. DC Subset Crosstalk: The Driving Force of the Anti-Tumor Response

Despite of the differential dysfunction of DC subtypes, the common fate is the failure of the anti-tumor immune response. Emerging data is suggesting a crosstalk among DC subsets in orchestrating the anti-tumor response, highlighting the role of each individual subset [123]. Figure 2 highlights some of this crosstalk as described below and the impact on the anti-tumor immune response. When DCs in the blood and tumor are perturbed, new interactions are created. This has been shown to occur between pDCs and cDC1s in the TME of patients with melanoma. These newly generated cross-interactions seems to be responsible for melanoma immune-escape [123]. cDC1 are central to this crosstalk by capturing and cross-presenting either MHCI and MHC II-restricted tumor antigens to CD4^+^ and CD8^+^ T cells [275]. In turn, activated CD4^+^ T cells are able to license cDC1s to drive CD8^+^ T cells cytotoxic responses, however many studies have questioned their role by reporting their dependence on other DC subset [275,276,277].

Contrastingly, Noubade et al. presented a model of anti-tumor immunity mediated by inter-DCs’ synergistic cooperation (for details see review) [275]. According to their scenario, intra-tumoral cDC1s and cDC2s capture TAAs [215] and migrate to the lymph node via CCR7 [177]. In the lymph node, cDC1s prime and activate naïve CD8^+^ T cells to produce CCL3/CCL4 and XCL1 to mediate the recruitment of CCR5+ pDCs and resident XCXR1^+^ cDC1s [215,278]. At the same time, cDC2 can cross-present TAAs to either CD4^+^ T and CD8^+^ T cells [203]. Once activated, CD4^+^ T cells induce the upregulation of costimulatory molecules (CD70, CD80, CD86) and cytokine (IL-12, IL-15) production the cDC1 to prime more CD8^+^ T cells. Alternatively, CD4^+^ T cells can directly exert anti-tumor responses by recruiting NK cells and macrophages through IFN-γ or direct cytolytic effects [279,280]. pDCs recruited to the tumor promote the maturation of infiltrating or resident cDC1s and activate CD4^+^ T and CD8^+^ T cell responses. In return, cDC1s secrete IFNλ [228] and IL-15 which promotes CD8^+^ T cell proliferation [281]. Activated pDCs also express high levels of CXCR3 ligands and CCR5 which attract more cytolytic lymphocytes into the tumor [282].

Inside the tumor, resident cDC1s are the primary source of CXCL9 and CXCL10 which modulate the recruitment of effector T cells [169]. NK cells are also present, which through the release of XCL1 and FLt3 induce cDC1 proliferation and accumulation [275]. Monocytes are also recruited intratumorally via CSF-1 and CCL2, with a pro-inflammatory phenotype and might differentiate into moDCs under the direction of Flt3 and GM-CSFR factors [138]. MoDCs help to amplify the anti-tumor response by TNF-α and IL-12 production [283,284]. Additionally, mregDCs might further contribute to activate antitumor CD4^+^ T cells, CD8^+^ T cells or NK cell responses via IL-12 production upon sensing IFN-γ released by neighboring T or NK cells [149].

### 5.3. Peripheral Blood DC Subsets (PBDCs)

A deep characterization of DCs in tumors allows us to predict their survival or response to immunotherapies. Despite that, the best picture of the TME comes from the immune landscape paired with the systemic characterization of circulating peripheral blood DCs (PBDCs). PBDCs exert a key role in shaping anti-tumor responses by continuously replenishing the pool of tissue resident DCs. Analysis of the peripheral cellular immunome can be also used to stratify individuals into different disease stages or to predict individualized clinical prognoses and treatment efficacy. Preoperative circulating myeloid DC levels in patients with pancreatic cancer have been considered a prognostic factor after surgical resection [285]. In addition, as seen TIDCs, frequencies of PBDCs have a distinct impact on clinical outcome depending on the DC subset. In melanoma patients, higher frequencies of circulatory cDC1s and pDCs positively correlate with progression-free survival, whereas high frequencies of cDC2s is associated with poor outcome [214].

TIDC profiling is more challenging due to the scarcity of DC infiltration, reduced sample size available for analysis, intra-tumor heterogeneity and complexity of DC subsets in the TME [286]. The accessibility of minimally invasive blood collections for PBDC characterization by flow cytometry, CyTOF, gene expression profiling and single cell sequencing, allows the simultaneous identification of several DC populations. This aspect is especially relevant for longitudinal analysis in clinical trials, which require repetitive immune-profiling analysis at different time points.

As with TIDCs, the frequency and distribution of PBDCs subsets varies based upon tumor type and other patient characteristics. The percentage and number of circulating cDC1s is significantly higher in patients with gastric cancer, compared with healthy controls [287]. Inversely, patients with pancreatic, prostate and NSCLC have been shown to have lower frequency of circulating cDC1s with impaired ability to stimulate T cells [168,214,288,289,290]. Meanwhile, cDC1 frequency in the blood is almost completely lost in patients with ovarian cancer [167]. Similarly, the cDC1/cDC2 ratio is decreased in advanced breast cancer compared with healthy controls [214,291,292]. A lower PBDC frequency may be explained by a negative influence of cancer cells on the generation of new DCs and by attracting them to the tumor through chemokine gradients, acting as a ‘black hole’ to inactivate them. Circulating CD1c^+^ DCs present with abnormal maturation defined as high expression of CD83 and/or CD86, low CD40, downregulation of the IFN-γR and impaired IL-12 secretory ability [214,293]. In melanoma patients, peripheral cDC1s have a defective ability to upregulate CD83 after TLR3/7/8 agonists stimulation ex vivo [294]. In breast, head and neck, and lung cancer, the inhibitory effects of VEGF on DC differentiation have been described to cause a drop in PBDCs, associated with accumulation of immature myeloid cells [295].

The frequency of circulating cDC2s has shown no alterations in patients with ovarian cancer or prostate cancer when compared to healthy donors. However, glioblastoma multiforme and breast cancer are characterized by a reduction in circulating cDC2s and pDCs compared to healthy controls [214,296]. PBDC maturity seems to be affected in some histologies, with increased expression of CD80, CD83 and CD86, principally in patients with ovarian cancer and a reduced CD40 expression in patients with prostate cancer [167]. In addition, cDC2 states associated with impaired interleukin IL-12 secretion, reduced IL-12/IL-10 ratio, low HLA-DR, CD86 expression and high CD40 expression have also be described [214,296].

No changes in circulating pDC populations were observed in prostate, oral squamous cell and squamous cell carcinoma of the head and neck [167,240,297]. However, melanoma and gastric cancer patients have shown an increased average of pDCs in circulation [255,298]. On the other hand, an accumulation of pDCs in malignant ascites with depletion in the blood has been described in patients with ovarian cancer with partial restoration reported after achieving complete remission through chemotherapy [239,299,300]. This decrease in circulating pDCs has also been described among different hematological malignancies [301,302,303] and in breast cancer at later stages [214,288]. However, a recent study reported differential frequencies in circulation related to HER-2 status with higher counts of periphereal pDCs in patients with HER-2 positive breast cancer compared to patients with HER-2 negative [304]. Many studies suggest circulating pDCs are positive prognostic factors in breast and pancreatic cancer patients of all ages, contrary to their presence in the tumor where they are associated with poor outcome [288,291].

Nevertheless, biomarkers based on circulating immune cells might not necessarily reflect immune cell composition in tumor [305]. This has been demonstrated by single-cell-RNA analysis reporting an increased phenotypic heterogeneity and expansion of myeloid lineage populations in breast cancer compared to normal tissue samples [305]. Although almost undetectable in the blood,14 novel myeloid clusters have been identified to be unique to tumor samples [305]. Certainly, the TME might contain heterogenous populations of inflammatory DCs and diverse local inflammatory microenvironment signals and many chemokines and cytokines can be secreted at much higher levels in the tumor tissue compared to blood [123,138]. Very few studies perform a direct comparison of TIDCs with PBDCs from the same patients [139,187,214,239,255,305,306,307]. With the advent of newer profiling technologies, a comprehensive and comparative analysis of all newly categorized DC subsets (DC4, DC5 and mreg DCs, etc), within TME and blood is required across tumor types to provide identification of markers or pathways that could be targeted in cancer.

## 6. Targetability of DCs in Cancer

In order to effectively target DCs in cancer, we must consider the multiple features that may be impeded by the tumor directly or the TME. We have categorized these features into 5 major areas: (1) Maturation and differentiation, (2) Ag processing and presentation, (3) cytokine production and migration, (4) immunomodulation of the TME, and (5) costimulation. As described below and illustrated in Figure 3, there are multiple approaches to overcome these suppressive avenues and drive anti-tumor DC properties.

### 6.1. Immunogenic Cell Death and DC Activation Induced by Radiotherapy and Chemotherapy

Existent cancer therapies can trigger DCs indirectly through their recruitment intratumorally or directly with the stimulation of signaling pathways, activation or inhibition by immune checkpoint or even DC-based adoptive cell therapy. Under indirect methods such as radiation, chemotherapy or physicochemical therapies, immunogenic cell death (ICD) is induced [308]. As a result of radiotherapy and chemotherapy, several DAMPs can be released and induce DC activation, intake of apoptotic bodies and the production of IL-6 and TNF-α [309,310]. In this context, DC chemotaxis to the TME is activated by the ATP released by tumor cell death recognized through purinergic receptors and NLRP3 resulting in IL-1β and IL-18 production [311,312,313]. Through ICD signaling, the tumor cell can produce IFN type I-II, CXCL10, HMGB1 and annexin A1 that all trigger DC maturation [314,315,316,317]. TLR3 and TLR4 ligation by the recognition of UVB-damaged RNAs and HMGB1, respectively, can mediate cytokine production and pre-activation of the inflammasome for IL-1β production [318,319].

A recent work conducted by Vanpouille-Box et al. showed that the DNA exonuclease Trex1 is crucial for proper DC activation induced by radiotherapy and that 12–18 Gy radiation is sufficient to upregulate Trex1 [320]. Radiotherapy can also activate the adaptor protein stimulator of IFN genes (STING) leading to IFN-α/β production by cancer cells [321,322]. Moreover, complement anaphylatoxins C3a and C5a released during radiotherapy, contribute in the recruitment and activation of DCs [323]. Most of the studies report these responses applying conventional γ and X-ray; however, alpha particles can also trigger DAMP-mediated DC activation [324].

### 6.2. Inhibitory Molecules Regulating DCs

Therapies targeting inhibitory molecules are designed to counteract the dampened function of DCs by blocking pathways that diminish their activation. Some target molecules described in this section include STAT3, IDO and immune checkpoint blockade.

#### 6.2.1. STAT3 Inhibitors

STAT3 is a protein member of a family of 7 TFs: STAT1-STAT6 and interact with JAK to induce their signaling pathway [325]. It can be activated by the IL-6 family, G-protein coupled receptors (GPCRs), growth factor receptors and TLRs [326]. STAT3 is linked to tumor growth and survival, as evidenced by STAT3 hyperactivation in several cancer types and as a poor prognosis marker. In breast cancer, it is strongly related to estrogen receptor α and drives metastatic disease in ER^+^ tumors with a more aggressive phenotype [327,328]. The TME can be modulated by STAT3 promoting the suppression of IFN--α/β, IL-12, TNF-α, CCL5, CXCL10 and upregulation of IL-6, IL-10, TGFβ and VEGF [325]. Deng et al. demonstrated that STAT3 specific expression by CD11b^+^ myeloid cells drives metastasis progression and its ablation in these cells reduced preformed metastatic niches [329]. Additionally, the presence of tumor-derived factors is associated with immature CD11b^+^ cell accumulation and a decrease in mature cells [330].

STAT3 inhibition can be accomplished using peptides, small molecules and oligonucleotides. Peptide-based strategies are directed to target different domains of STAT3 to prevent dimerization. Interestingly, the selective JAK2/STAT3 inhibitor JSI-I24 can revert the tumor-derived factors that block DC differentiation and maturation inducing the upregulation of MHC class II and costimulatory molecules [331]. In melanoma and primary glioma, the blockade of STAT3 and p38 MAPK signaling pathways promoted the differentiation of CD14^+^ myeloid cells to CD1a^+^ [332]. STAT3 blockade also can revert DC immunosuppression by inducing maturation, thus is associated with an increase in tumor infiltrating CD4^+^ and CD8^+^ T cells and a decrease in T regulatory cells.

#### 6.2.2. IDO Inhibitors

Several cell types constitutively express IDO including bone marrow, muscle, gastrointestinal tract, kidney, APCs, and B cells among others [333]. While IDO1 and TDO can be found in the normal liver and lung, a recent study compared expression amongst 31 cancer types with normal tissue equivalents and found that expression was highest in the tumor tissue in all cases except liver and lung [334]. IDO expression has a controversial impact in the TME with higher IDO expression correlated with a decrease in tumor growth and longer survival in patients with prostate cancer [335], hepatocellular carcinoma [336] and renal cell carcinoma [337]. Conversely, IDO is often related to tumor progression, formation of metastatic niches in lung carcinoma and breast carcinoma-derived pulmonary metastasis [338], a reduction in T cells and a higher frequency of liver metastasis in colorectal cancer [339] and reduced TIL and NK cells in endometrial carcinoma [340].

There is a strong relationship in the induction of IDO transcription with proinflammatory cytokines like TNF-α, IL-6, IL-1β, IFN-γ as well as CTLA-4 and costimulatory signaling pathways that increase IDO expression in DCs [341,342,343,344,345]. Moreover, IDO^+^ pDCs are found in tumor draining lymph nodes and correlated with poor prognosis and metastasis [346]. Wang et al. described the effect as bystander suppression, in which the IDO^+^ DCs suppress T cell response to antigens presented by nearby IDO negative DCs [347]. IDO inhibition has been demonstrated to revert the immunosuppression and T cell exhaustion effect. In DCs, IDO can be induced in vitro by prostaglandin E2 and displays a potent inhibition effect on allogenic T cells that can be restored with its inhibition [348]. The competitive inhibitor 1-methyl-tryptophan (1-MT) in AML abrogates Treg induction by IDO^+^ tumor cells and promotes the expansion of CD4^+^CD25 T cells [349]. The testing of an IDO-silenced DC vaccine in a small cohort of patients with ovarian and uterine cancer demonstrated that DCs upregulate CD40, CCR7 and may enhance their immunogenic functions in the absence of IDO while CD80 and CD86 do not appear affected [350]. Conversely, a synergized effect of 1-MT in combination with radiotherapy can upregulate CD80, CD86, MHC class II in DCs [351]. Brown et al. revealed that the combination of indoximod and anti-CTLA-4 overcame resistance to immune checkpoint blockade [352]. This was also observed with the PF-06840003 inhibitor that showed a resulting reversal of T cell exhaustion in combination with anti-PDL1 [353]. Recent phase I/II/III clinical trials have shown that combination of immune checkpoint blockade therapy or radiotherapy with IDO1 inhibitors seem to be more promising than their use alone [354]. However, to date, no phase III study blocking this pathway has shown a significantly improved patient outcome.

#### 6.2.3. Immune Checkpoint Blockade

##### PD-1/PD-L1 Axis

Amplification of the antitumoral response by blocking co-inhibitory receptors like PD1, PD-L1 or PD-L2 and CTLA-4 has resulted in a sustained, positive clinical outcome in several cancer types [355]. Often PD-L1 expression is associated with Th1 and CD8^+^ T cells and can be upregulated by IFN-γ or by IFN-β [356,357]. Interestingly, the tumor cells modulate their PD-L1 expression to avoid the CTL-driven antitumoral response and corrupt the discriminatory functions of the immune response as an evasion mechanism [358]. However, ligand binding is also described in cis thus interfering with the interaction of PD-L1 to PD-1 or B7.1 to CD28 on T cells [359]. The final effect is an impaired DC costimulatory signal that induces a defective anti-tumoral T cell response and resistance to anti-PD-1 therapy. The same study reported that DC1 and DC2 populations express higher PD-L1 than B7.1 with marginal expression of PD-1 on DC1 as compared to pDCs who only expressed PD-L1 [193]. Currently developed blocking strategies for this axis include antibody blockade, gene silencing and small-molecule pathway inhibition [360]. Generally, blockade of the axis with either anti-PD-L1 or anti-PD-1 prolongs survival with variable response rates across different cancer types. A recent study in a murine model of melanoma showed that tissue resident CD103^+^ DCs are needed for the generation of anti-tumor effector T cell responses through TAA delivery to tumor-draining lymph nodes during PD-L1 blockade [172]. Additionally, the therapeutic effect induced by PD-L1 blockade disappears in a DC-conditional PD-L1 knockout setting even though other cells continued to express the ligand thus demonstrating a unique role of the DC in the success of this therapy. Observations in patients with asymptomatic multiple myeloma undergoing anti-PD-L1 therapy (atezolizumab) showed inflammasome activation, increased expansion of circulating myeloid cells, and CD40L-driven mature DCs. Conversely, treatment with anti-PD-1 therapy in this study did not result in the same inflammatory signature [361].

On the other hand, the success of anti-PD-1 therapy has been linked to TIDCs and IL-12 production despite of the inability to directly bind the DCs. Removal of this inhibitory signal on T cells results in renewed production of IFN-γ which, in turn, induces NF-κB activation for IL-12 transcription by DCs, demonstrating a key role of T cell-DC crosstalk in mediating therapeutic responses [263]. DC dependent responses to PD-1 blockade have been associated with BATF3-dependent DCs. In a murine model, loss of these cells resulted in a reduced priming of TAA-specific CTLs following anti-PD-1 or anti-CD137 treatment [362]. The FDA approved antagonistic anti-PD-1 blocking antibodies are nivolumab for use in urothelial carcinoma, NSCLC, melanoma, renal cell carcinoma, head and neck squamous cell cancer; pembrolizumab for NSCLC, melanoma, Hodgkin’s lymphoma, cervical and gastric cancer and cemiplimab for cutaneous squamous cell carcinoma. Despite success in some tumor types, others have shown a minority of patients experiencing durable responses with some histologies showing an almost completely refractory outcome to checkpoint blockade such as colorectal and ovarian cancer [358,363].

The combination of PD-1 blockade with other therapeutic strategies such as radiotherapy, neoadjuvant therapy, and IL-2-based cytokine approaches increase complete responses, overall survival, and antitumor responses mediated by CD8^+^ T cells and CD20^+^ B cells. This has been demonstrated in renal cell carcinoma, NSCLC and metastatic melanoma [358,364,365,366,367]. Recently, in vitro treatment of an osteosarcoma cell line with tyrosine kinase inhibitors in combination with nivolumab lead to DC maturation and polarization of IFN-γ-secreting effector T cells as well as a reduction in Tregs [368]. Combination therapies utilizing PD-L1 blockade have also shown a synergistic effect. The use of DC vaccination in combination with pomalidomide and PD-L1 blockade in a multiple myeloma model reduced tumor growth [369]. In a pancreatic cancer model, the combination of CCL21, a chemoattractant for DC and T cells, and PD-L1 blockade resulted in a synergistic tumor suppression [370].

##### CTLA-4 Inhibition

Blockade of this checkpoint receptor allows for the priming of naïve T cells in lymphoid organs. Since the discovery of the role of CTLA-4 in the TME, several inhibitors have been developed [371]. The proposed mechanism for restoration of the antitumor immune response when blocking CTLA-4 is through removing the negative regulation of effector CD8^+^ T cells and reducing intratumoral Treg populations [372]. Two antagonistic anti-CTLA-4 antibodies used clinically are tremelimumab and ipilimumab. Long-lasting response rates in melanoma have been reported for 6–20% of patients but mixed results exist regarding a reduction in Tregs with treatment [373,374,375,376,377,378]. It has been shown that CTLA-4 can also be expressed on breast cancer cells and induces a suppressive effect on DCs by downregulation of costimulatory markers and HLA-DR. This effect can be reversed by CTLA-4 blockade. The restored DCs mature and produce normal cytokines and prime Th1/CTL responses [379].

A direct effect on DCs is achieved by the combination of anti-CTLA-4 blockade with other types of therapies which have shown promising responses in murine models. A recent study demonstrated a synergistic effect when CTLA-4 was downregulated in tumors though siRNA targeting, and DC vaccination was used to potentiate T cell priming. Additionally, they observed a reduction in the frequency of immunosuppressive cells, increased CTL activation and proinflammatory cytokines, and a reduction in angiogenesis and metastasis [380]. Comparable results were obtained in colon cancer model when the tumor was irradiated and immature DCs were adoptively transferred in combination with anti-CTLA-4. They showed an increase in survival rate, CTL activation and IFN-γ secreting T cells [381]. In all the cases described, the combinatory effect leads to the maturation and correct function of intratumoral DCs.

### 6.3. Molecules for DC Activation

#### 6.3.1. TLR Agonists

The TLR signaling pathways triggered by different PAMPs or DAMPs can induce the production of proinflammatory cytokines including IFN-α/β. This feature has been studied as an adjuvant mechanism for modulation of the TME to enhance the anti-tumoral effects mediated by intratumoral myeloid or T cells that express the target TLR(s) [382]. Additionally, polymorphisms on the receptors or pathway molecules have been associated with an increased risk for several cancer types such as colorectal carcinoma [383], hepatocellular carcinoma, NSCLC [384], breast carcinoma [385], cervical cancer [386], and oral squamous cell carcinoma [387]. TLR agonists have also be used to elicit an adjuvant effect in therapeutic tumor vaccination studies with TLR3/7/8/9 being the most targeted. Over the past few years, works attempting this effect demonstrated that TLR stimulation induces DC maturation, reduces phagocytic capacity and increases costimulatory molecule expression, and favors Th1 polarization and production of cytokines like IL-12 and IFN-γ [388,389,390]. Administration of a TLR3 agonist, polyI:C, induces therapeutic activity in melanoma [391], colorectal cancer and breast cancer in preclinical models [392]. The combinatory effect of TLR3 agonists with other therapies improved the magnitude and duration of the anti-tumoral responses. CD103^+^ DC progenitors activate and expand through the systemic administration of FLT3L followed by intratumoral polyI:C injections with the administration of immune checkpoint blockade enhancing the anti-tumoral effect and limiting tumor growth [171]. Similarly, administration of CpG oligo dinucleotides (ODN), a TLR9 agonist, primes tumor-draining lymph node DCs that produce IFN-α/β resulting in T cell activation. Currently several preclinical and phase I/II studies utilizing adjuvant therapy triggering TLR9 in melanoma, glioblastoma, colorectal cancer, NSCLC and others are ongoing [393]. A combined administration of TLR7 and TLR9 agonists with PD-1 blockade in head and NSCC suppresses tumor growth with an abscopal effect observed related to TAM and CD8^+^ T cells activation [394]. pDCs stimulated with CpG ODN attain a mature phenotype and a high secretion of IFN-α [395]. Furthermore, intratumoral administration the TLR7/8 agonist, MEDI9197, also favors Th1 polarization and IFN-γ production and synergizes with immune checkpoint blockade [396]. TLR4 stimulation has be shown to result in the secretion of TNF-α that, in turn, increases the ability of DCs to mature and migrate, prime Th1 responses and cross-present to CD8^+^ T cells [397,398]. Differential DC subset activation depends on the specific expression profile of TLRs across these subsets and across maturation states. However, targeting TLRs has been shown to have the most therapeutic potential when combined with other strategies such as immune checkpoint blockade or DC vaccination.

#### 6.3.2. STING

The signaling pathways of the recognition molecules sensing DAMPs can be also targeted to potentiate the DC response. That is the case of the Stimulator of Interferon Genes (STING), intimately related to the cytoplasmatic sensor cGAS [399]. The essential role of STING is to produce IFN-α/β in response to both exogenous or endogenous cytosolic DNA (cDNA) [400]. This pathway is independent of JAK-STAT; however, it is a potent recruiter of IRF3 to induce IFN-α/β. It is widely distributed in immune and non-immune cells and is member of the ER transmembrane protein family [401]. In cancer, IFN-α/β can impair immunity as IFN-β is associated with the increase in PD-L1 and PD-L2 expression by tumor cells [402]. Additionally, it has been related to chemotherapy and radiotherapy resistance and the occurrence of autoimmune toxicity during immunotherapy-based therapeutic strategies. Therefore, STING is necessary for anti-cancer immunity, including the promotion of DC maturation and activation through IFN-α/β [403]. Recently it has been reported that the expression of STING in non-tumor cells promotes NK tumor cytotoxicity, apoptosis of malignant B cells and remodeling of the tumor microenvironment by reduction in MDSCs in nasopharyngeal carcinoma [404,405,406]. In DCs, by promoting IFN-α/β dependent activation, STING enhances the presentation of TAAs and cross presentation to CD8^+^ T cells [403,407]. The cyclic dinucleotides (CDNs) are principal mediators of this signaling pathway and synthetic CDNs have been produced as STING agonists. Intratumoral injection of CDNs in melanoma and colorectal cancer mouse models demonstrate tumor regression and systemic anti-tumor T cell immune responses [408]. The combination with a GM-CSF tumor vaccine in preclinical models also had shown activation of DCs and anti-tumor specific CD8^+^ IFNγ^+^ T cells [409]. Interestingly, stimulation of the STING pathway through damage response proteins induces higher expression of PD-L1 and potentiates ICB therapy in SCLC [410]. This enhanced effect was also demonstrated in murine melanoma models by the combined administration of cGAMP and anti-PD-L1 [411]. Ongoing clinical trials in metastatic HNC and lymphoma, solid tumors and NSCLC are testing the use of CDNs combined with ICB therapy or are used as adjuvants for chemotherapy [406,412]. Regardless of the beneficial therapeutic effects, important limitations remain to be overcome such as the intratumoral delivery system efficacy and the short half-life of CDNs in circulation due to their high rate of degradation; novel drug delivery options need to be explored.

#### 6.3.3. CD40 Agonist

This co-stimulatory member of the TNF superfamily is expressed by immune and non-immune cells. DCs express CD40 which is associated with maturation and an activation phenotype necessary for T cell priming, but also promotes DC survival and an anti-tumor response [413]. In contrast, the lack of CD40 and IL-23p19 is related to DC tolerogenic programs [414]. Analysis of TCGA data of CD40 expression in pancreatic adenocarcinoma, melanoma and RCC demonstrated a strong correlation with Ag presentation molecules, TLRs and T cell function supporting the use of CD40 agonists as an attractive cancer immune therapy for DC activation [415]. Preclinical models using agonistic anti-CD40 mAb have shown secondary T cell activation and tumor regression [416,417]. In patients with PDAC, on-treatment biopsies revealed low fibrosis, a higher density of mature DCs, CD4^+^ and CD8^+^ T cells, and the coadministration of chemotherapy induced the greatest intratumoral responses [418]. Additional studies have assessed a combination of RT and CD40 activation therapy finding an increased response to ICB, inducing a complete remodeling of the microenvironment including upregulation of MHC class I, costimulatory molecules, activation of DCs and infiltration of CD8^+^ T cells [419]. It has been related to a reversal of resistance to anti-PD1 therapy by the down regulation of PD-1 on T cells, and reduction in CTLA4 and PD-1 on Tregs [420]. The same effect was found in patients with osteosarcoma in which the efficacy of anti-PD1 therapy was enhanced [421]. A case report of a patient with melanoma that attained completed remission following anti-CD40 treatment demonstrated the expansion of a de novo TCR repertoire identified years after completion of the therapy suggesting a durable impact on the immune response [422]. It is important to note that single agent anti-CD40 agonist has shown no clinical or minimal clinical responses in patients and this therapy should be conducted in a combination setting [423,424].

## 7. Moving towards a DC—Based Personalized Cancer Therapy

Many therapeutic strategies to potentiate DC driven anti-tumor immunity have been proposed, including immune-checkpoint blockade, depletion of immunosuppressive DC subsets, and antibody-mediated or vaccine-mediated DC activation. Despite that, the majority fail in their efficacy or in providing contrasting results when assessed in clinical trials.

Currently our “incomplete” understanding of the TME, myeloid cell heterogeneity and cell–cell cross talk in solid tumors has led to the movement of several strategies into the clinic prior to careful target selection based upon rational mechanistic hypotheses [207]. The picture is made even worse by the innate multi-state variability of immune cells, which can dynamically shift among different maturation and activation stages. As such, DC targetability in cancer is still in its infancy. Research is needed to focus on ways to optimally exploit specific DC subsets with specialized functions in order to prime an efficacious anti-tumoral response. We have highlighted 5 main areas to consider as well as a proposed workflow in Figure 4: (1) the incomplete and non-standardized classification of the subsets at steady state, which complicates the phenotyping and functional assessment in healthy conditions and across different tumor types, (2) the limited analysis of circulating DCs from patients with cancer and their impact and contribution to the TME, (3) deepening our understanding of DC crosstalk both between different DC subsets and other immune cells in the TME that could enhance activation or inhibition of the cells, (4) large scale computational analysis with the ability of large scale scanning and finally (5) the development of novel drug delivery systems that trigger DCs within the tumor.

As highlighted in the previous sections, DC subsets can exert different functional behaviors in the TME. As such it is essential to evaluate TIDC presence prior to therapy to refine the appropriate method to target them and elicit the relevant immune response. Indeed, cDC2 and cDC1 vaccinations have been shown to fail in tumors with a paucity of myeloid cells or in tumors enriched with M2 macrophages, respectively [138]. Recently there has been an exponential increase in studies applying single cell genomic and transcriptomic methodologies with the goal of defining specific molecular profiles of DC subsets among different tumor types. scRNA seq is providing a glimpse into DC phenotype diversity, heterogeneity of DC subset abundance in the TME, ontogeny and differentiation. Transcriptomic comparison between tumor and uninvolved juxta-tumoral tissue have identified specific ‘imprinting” on individual DC subsets dictated by the effects of the TME and their interactions with other tumor-associated cells [201]. Transcription factor–target modeling and pathway–target modeling are useful tools that allow for the detection of upregulated signaling pathways related to DC-specific gene networks, which are helpful for subtype classification and targetability among different cancers. For example, data considering three different tumor types reveled gene upregulation associated with ROS and NOTCH signaling and keratan sulfate metabolism in tumor infiltrating cDC1s; hypoxia and Wnt/β catenin pathway and methionine metabolism were found in cDC2s; TNFα/NFκB and IFN signaling, together with folate and tryptophan metabolism in migratory cDC2s; and Kras signaling and sphingolipid, lysine, purine and arginine metabolism in pDCs [201]. Therefore, since cDC2 are the highest frequency subtype inside these tumors, inhibition of the hypoxia and Wnt β catenin pathway might be a valid therapeutic approach. In the same direction, Michea et al. have identified pDCs as the most infiltrated subtype with the highest gene signature dysregulation in the breast cancer TME. Among them, the “anatomical structure involved in morphogenesis” (EPHB1, VEGFB and VASH2) pathway represents the most significantly enriched pathway, hence, the potential for a hypothetical target of this pathway [166].

Despite that, scRNA seq analysis is not enough. Challenges in the interpretation of scRNA seq data to identify new DC populations were highlighted by Villar et al. [114], suggesting the requirement of high-dimensional multi-omics analysis. Lavin’s study has proposed a methodological approach based on a barcoding strategy which allows for the simultaneous measurement of single-cell expression by CyTOF analysis combining with single-cell transcriptomics of immune cells residing in blood and tumor sites for each patient. This significantly avoids any batch effect and technical variability inherent to sample processing among blood and tumor samples and, taking advantage of the large spectrum of markers that can be identified, allows for the identification of multiple DC subsets in a single assay [187]. As previously mentioned, the functional and phenotypic status of DCs significantly changes based upon location (tumor vs. blood). One such example is the study by Sosa Cuevas et al., where the authors assessed DC features in the blood and in the tumor. They found that functional and phenotypic differences existed based upon location and that the proportion of cDC1s in the blood was the most accurate parameter to predict clinical outcome of melanoma patients [214].

The crosstalk between DC subsets and other immune cells within the TME can alter biological outcomes and the anti-tumor response. Given their importance in antigen cross-presentation, cDC1s have been delineated as a privileged target subset for improving anti-cancer immunity through their cooperation with other immune subtypes [275]. Therefore, targeting maturation signals specific to just cDC1s may be not the optimal approach. A multi-target therapeutic strategy, which delivers signals able to enhance the function of multiple DC subsets may be needed to enhance efficacy and durability of the response [275]. In this regard, new computational methods are emerging with the ability to predict receptor-ligand binding between interacting cells through combining expression data with prior knowledge of signaling and gene regulatory networks [425].

Following the generation of large-scale Omics, the next step is to computationally analyze all the identified biomarkers, frequency of DC subsets and interactions with other immune cell types, clinical information, and epidemiological data. This will allow researchers to evaluate the overlap of the characterized DC status between patients and cancer types, to assess their association with survival and, hence, to provide a precise patient stratification to guide therapeutic strategies. Finally, many DC-targeting therapies are administered intravenously which may result in impaired efficacy and side effects. Gene therapies, small molecule inhibitors or other targeted approaches that need a local release to mediate their function require a new tactic in drug delivery. Nanoparticle delivery systems represent a promising therapeutic approach that may provide an opportunity for a multi-target delivery of co-stimulatory molecules, soluble mediators, and DC specific receptors thus reviving the anti-tumor immune response.

While DC specific therapies have faced several challenges and hurdles, we are now entering an exciting time of bedside to bench and back to bedside discoveries. Following this roadmap, in the future we will be able to create a tumor DC-specific atlas which represents a valuable resource to identify clinically relevant phenotypes, crosstalk, biomarkers for patient stratification and precision medicine applications that may be linked to a more directed drug delivery system and improved clinical outcome.

## Figures and Tables

**Figure 1 cells-11-03028-f001:**
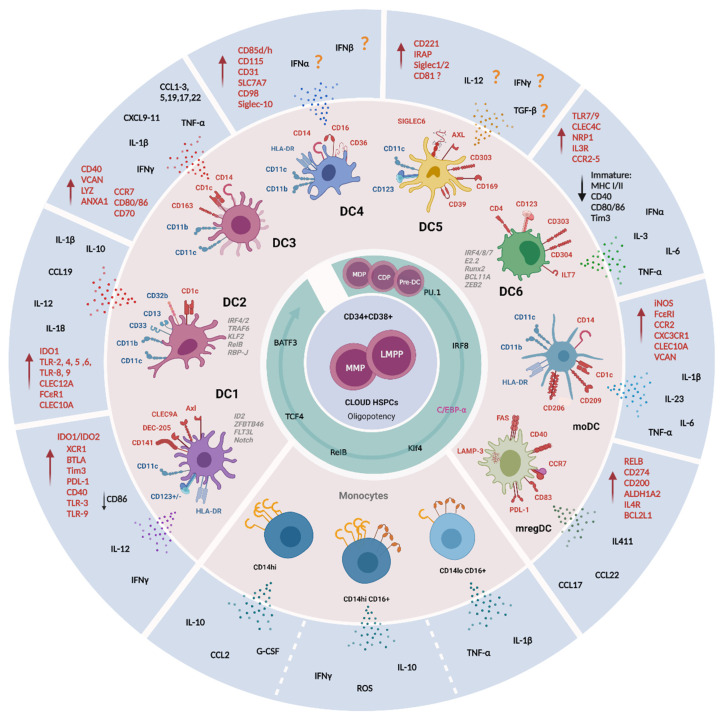
Overview of DC and monocyte subsets. The inner circle shows key differentiation associated TFs and proteins. The second circle shows the subsets with canonical TFs, surface and intracellular markers, the red color markers are the key molecules to distinguish the subpopulation, the blue color shared markers between populations, main transcription factors for some subpopulations are described in gray. The outer circle illustrates secreted proteins, surface receptors and key protein changes upon activation, red arrows indicate upregulation of the molecules, the black arrow down regulation. The molecules with question marks are proposed based on the genomic profile found in the subpopulation.

**Figure 2 cells-11-03028-f002:**
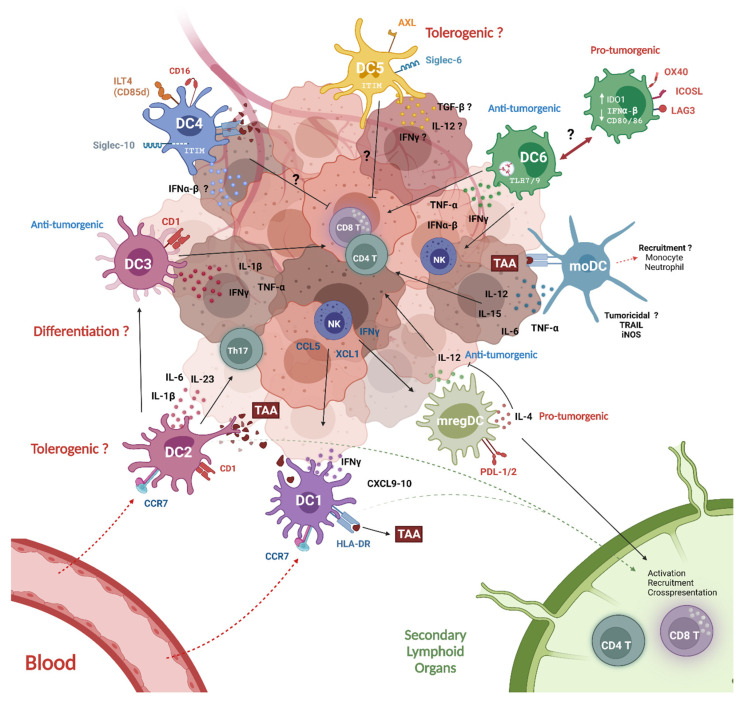
DC function and cross-talk within the tumor microenvironment. Proposed and validated functions of DCs and monocytes within the TME are shown. Proposed functions are indicated by the inclusion of a ‘?’ with the feature. Major cytokines, chemokines, and surface markers are indicated. Arrows show how the DC or monocyte subset may engage with other immune cells within the microenvironment.

**Figure 3 cells-11-03028-f003:**
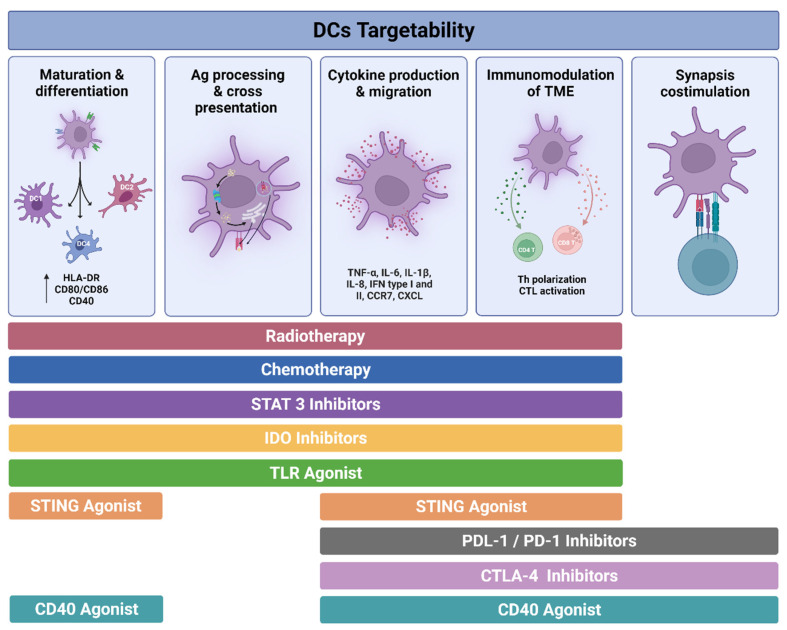
Strategies to target DCs in cancer. Five mechanisms are shown that can be targeted by various therapeutic approaches that are shown in the rows below. This may not be inclusive of all targetable mechanisms or potential therapeutic strategies.

**Figure 4 cells-11-03028-f004:**
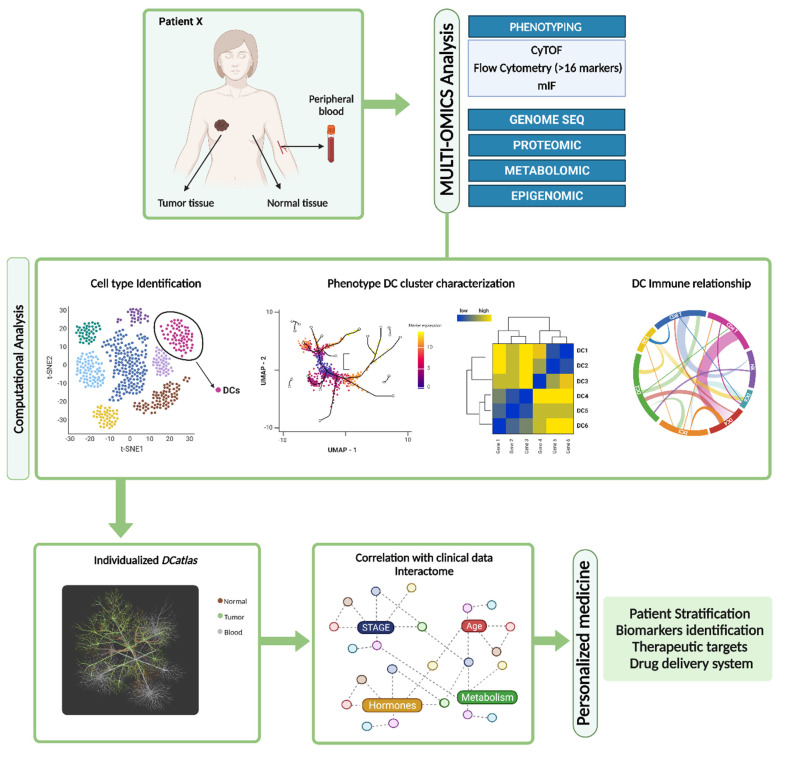
Proposed workflow to expand our understanding of DC biology, functional role and targetability in cancer.

## Data Availability

Not applicable.

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
