# Peer review of "Dendritic Cells: The Long and Evolving Road towards Successful Targetability in Cancer"

_cells, 2022, doi:10.3390/cells11193028_

Round 1

Reviewer 1 Report

Dendritic cells are key cells of the immune system and therefore data on their functioning and different interactions are fundamental for finding adequate new treatments for tumors treatment. 

Poor response to immunotherapies is leading researchers to try to target specific cells, such as tumor Dendritic cells, in order to induce an anti-tumor immune response.

The review topic is widely covered and the topic of the manuscript is of great relevance and gives the possibility to go  through the entangled network existing around dendritic cells. 

Tabes and figures are well presented amd result vey explicable.

Author Response

Thank you for the feedback on our review article! We appreciate your time and comments. 

Reviewer 2 Report

This review aims to provide an updated overview on Dendritic cells (DCs), a unique myeloid cell lineage that play a major role in both adaptive immune response and tumor microenvironment (TME). The topic is clinically relevant since DC are likely to be an attractive target for immune oncology based therapeutic approaches.

The review provides a deep descriptive background summarizing state of art on DCs and cancer. It describes in detail the advances in DC origin, subset classification, function, DC crosstalk within the TME and finally targetability in homeostasis and cancer moving towards a DC-based personalized cancer therapy discussion.

The review is fluent and easy to read. It is also well written. The figures and the table are well-made and describe in detail what is reported in the text. Text presents some minor concerns that need to be revised.

Minor concerns:

1.      Lane 134: “cDC1” should be spelt out.

2.      Lane 238: “PD-L1” should be spelt out.

3.      Table should be better formatted. In addition, in the column refers to DC subpopulation, “(cDC1)” should be written “cDC1s”; the authors should add “or DC2” at “Classical Type 2 DCs (cDC2s)”, as well as at “Monocytes-derived DCs”, they should add “(mo-DCs)”.

4.      Lane 371: “HNSCC” should be spelt out.

5.      “IFN” word is repeated several time in the manuscript but each time it is reported in different form as “type I-IFN”, or “IFN 1”, or “IFN type 1” and other time the subtypes are not reported. For this reasons, the authors should verify in the text the appropriate and correct form.

6.      Lane 537: there is an empty bracket.

7.      The authors should correct the numerical order of paragraphs because after 6.1.5 is reported 6.3 and not 6.2 and consequentially they should change also 6.3 instead of 6.4.

The same error for the paragraph 7.2.3.1 that is repeated twice.

8.      Lane 768: “tumor microenvironment” should be written “TME” because already reported.

9.      Lane 867, 936, and 953: the IFN subtypes are not shown.

10.  Lane 1075, the word “TNF/NF” should be correct.

Author Response

Thank you for your constructive feedback on our article! We have adjusted the text according to all the comments below. For the table, we have adjusted the format and are happy to try to adjust again as needed.  

Reviewer 3 Report

In the manuscript titled Dendritic Cells: The long and evolving road towards successful targetability in cancer, the authors reviewed the origin, development, functions and new subsets of DCs, in addition, the crosstalk of DCs with TME and some target molecules for improving immunorepressive TME in DCs were talked. Totally, the author has systematacially introduced DCs, which promotes the design of new therapeutic strategies in cancer. But the basic introduction of DCs in different part are too redundant, some basic knowledge need reduced. And the achievable therapeutic strategies and the authors' suggestions should be raised and emphasized.

Author Response

We would like to thank the reviewer for their feedback and critique. We appreciate the time you spent reviewing our review article. 

  1. We have removed text that we felt was redundant or provided to much background that may not be necessary for the reader assuming a certain level of immunology background. This has been updated in the revised manuscript using track changes.
  2. We have added language to the abstract to emphasize some key points regarding our review. 

We look forward to any further feedback based upon our revised version. 

Round 2

Reviewer 3 Report

What I concerns have been dispelled.